# EGFR-Binding Peptides: From Computational Design towards Tumor-Targeting of Adeno-Associated Virus Capsids

**DOI:** 10.3390/ijms21249535

**Published:** 2020-12-15

**Authors:** Rebecca C. Feiner, Isabell Kemker, Lea Krutzke, Ellen Allmendinger, Daniel J. Mandell, Norbert Sewald, Stefan Kochanek, Kristian M. Müller

**Affiliations:** 1Cellular and Molecular Biotechnology, Faculty of Technology, Bielefeld University, 33615 Bielefeld, Germany; rebecca.feiner@uni-bielefeld.de; 2Organic and Bioorganic Chemistry, Faculty of Chemistry, Bielefeld University, 33615 Bielefeld, Germany; isabell.kemker@uni-bielefeld.de (I.K.); norbert.sewald@uni-bielefeld.de (N.S.); 3Department of Gene Therapy, Ulm University, 89081 Ulm, Germany; lea.krutzke@uni-ulm.de (L.K.); ellen.allmendinger@uni-ulm.de (E.A.); stefan.kochanek@uni-ulm.de (S.K.); 4Department of Bioengineering and Therapeutic Sciences, University of California San Francisco, San Francisco, CA 94158, USA; dan.mandell@grobio.com; 5Bioinformatics Graduate Program, University of California San Francisco, San Francisco, CA 94134, USA

**Keywords:** cyclic peptide, VDEPT, oncolytic virus, protein engineering, VP protein, synthetic biology

## Abstract

The epidermal growth factor receptor (EGFR) plays a central role in the progression of many solid tumors. We used this validated target to analyze the de novo design of EGFR-binding peptides and their application for the delivery of complex payloads via rational design of a viral vector. Peptides were computationally designed to interact with the EGFR dimerization interface. Two new peptides and a reference (EDA peptide) were chemically synthesized, and their binding ability characterized. Presentation of these peptides in each of the 60 capsid proteins of recombinant adeno-associated viruses (rAAV) via a genetic based loop insertion enabled targeting of EGFR overexpressing tumor cell lines. Furthermore, tissue distribution and tumor xenograft specificity were analyzed with systemic injection in chicken egg chorioallantoic membrane (CAM) assays. Complex correlations between the targeting of the synthetic peptides and the viral vectors to cells and in ovo were observed. Overall, these data demonstrate the potential of computational design in combination with rational capsid modification for viral vector targeting opening new avenues for viral vector delivery and specifically suicide gene therapy.

## 1. Introduction

The epidermal growth factor receptor (EGFR/ErbB1) is a member of the ErbB receptor family. ErbB receptors are well-studied receptor-tyrosine kinases (RTKs) that additionally comprise Her2 (ErbB2/NEU), Her3 (ErbB3), and Her4 (ErbB4) [1]. Structurally, the receptor consists of an *N*-terminal extracellular domain, a transmembrane domain and an intracellular domain [2,3]. The extracellular part can be subdivided into four domains (I to IV). Domains I and III mediate the contact to the ligand, while domain II is mainly involved in homo- and heterodimerization via the so-called dimerization arm [4]. Exposure of the dimerization arm is promoted when a ligand binds domains I and III and the receptor switches from the auto-tethered form to the extended conformation [5,6]. This extension enables receptor dimerization that initiates diverse intracellular signaling cascades for cellular processes including cell division, migration, adhesion, differentiation, and apoptosis [7,8,9]. Seven different ligands are known to bind and activate EGFR including the epidermal growth factor (EGF) [10]. Cellular response is strongly influenced by the ligand and its interaction with the receptor that induces either an asymmetric or a symmetric dimerization [11,12]. In healthy tissues, the availability of EGFR ligands is strictly limited in order to ensure tight control over signaling processes and cell proliferation [13]. In cancerous tissues the EGFR is often perpetually stimulated or the receptor itself is constitutively activated by mutations [14]. Alterations to the EGFR copy number are often observed in head and neck squamous cell, colorectal, squamous non-small-cell lung, and breast cancer [15,16,17,18]. Therapeutic inhibition of EGFR ligand-binding sites has been explored since the 1980s when abnormal expression and activation of EGFR was first observed [19]. Several monoclonal antibodies that compete with EGFR ligand binding have since gained regulatory approval including Cetuximab, Panitumumab, and Necitumumab [20,21,22]. However, tumor cells may develop mutations to the EGFR extracellular binding domain and gain resistance against those antibodies. The approved anti-ErB2 antibody Pertuzumab presents another mode of inhibition by binding to the membrane proximal dimerization interface, which sterically hinders the receptor-receptor interaction and thereby blocks downstream signaling [23,24]. This mechanism of action validates the dimerization surface as a target for the design of ErbB antagonists.

As an alternative to antibody therapy, peptides have also been evaluated for EGFR targeting. The small size of peptides in comparison to antibodies improves solid tumor penetration, and they can be chemically synthesized at a comparatively low cost. Peptides characterized for their EGFR binding include the 12-mer peptide GE11 (YHWYGYTPQNVI), which was identified in 2005 from a phage display peptide library [25] and the 6-mer peptide D4 (LARLLT), which was selected from a computer-aided design approach [26]. The latter was presented on liposome nanoparticles to provide specific targeting of cancer cells. Integration into larger structures of liposomes or virus-like particles offers the possibility to overcome common disadvantages of peptides like rapid in vivo renal clearance and low affinity. Combining several peptides on the surface of a nanoparticle can also increase therapeutic effects through avidity [27,28]. Other approaches in finding EGFR-binding peptides focused on targeting domain II and discovered the minimal binding motif of the so-called EGFR dimerization arm (QTPYYMNT). Subsequent work showed that this peptide interacts with EGFR and inhibits signaling [29,30,31].

In this work we developed peptides that bind the EGFR domain II dimerization arm to target adeno-associated virus (AAV) to EGFR overexpressing cells. AAVs are non-enveloped, non-pathogenic viruses, which harbor a single-stranded DNA genome of about 4.7 kb flanked by inverted terminal repeats (ITR), which provide the encapsidation signal [32]. By providing the replicative- and capsid-forming genes in trans, any gene of interest can be placed between the ITRs to yield recombinant adeno-associated viruses (rAAV). The genetic payload can for example code for a prodrug activating enzyme (e.g., thymidine kinase) to facilitate virus-directed enzyme prodrug therapy (VDEPT) [33]. Specific delivery of rAAVs to tumor cells poses a major challenge to gene therapy and different approaches have been evaluated to improve a tissue specific tropism [34]. To enable therapeutic targeting of AAV to tumor cells, we computationally designed peptides intended to bind the dimerization arm region in domain II of EGFR. Linear and cyclic versions of two peptides were synthesized and characterized for EGFR binding in cellular assays. Genetic incorporation of these peptides into the AAV structural VP proteins (VP1, VP2, and VP3) enabled the production of rAAV presenting the EGFR-binding peptide in all 60 proteins forming the capsid. We further abrogated the natural tropism by introducing the known VP mutations R585A and R588A that prevent binding to the natural primary viral receptor heparan sulfate proteoglycan (HSPG) and block native transduction [35,36,37]. The resulting rAAV transduced EGFR overexpressing tumor cells but not a control cell line. We further determined the initial tissue distribution of rAAV transduction by rapid in ovo analysis using chorioallantoic membrane (CAM) assays with a tumor xenograft.

## 2. Results

### 2.1. Computational Design of β-Hairpin Peptides Competitively Inhibiting Dimerization of EGFR

We aimed to engineer β-hairpins that competitively inhibit dimerization of EGFR by mimicking the EGFR β-hairpin dimerization arm (Figure 1A). We targeted the dimer interface because it presents a cavity amenable to peptide design, and furthermore is essential for receptor activation and therefore rarely affected by mutations [24]. We began by extracting the 3D structure of 200 12-residue β-hairpins from the protein databank (PDB) that are closest to type I or type II β-hairpins. We set the initial coordinates for the design simulations by superimposing these structures onto the native hairpin of the EGFR dimerization arm. In the next step, each structure was rotationally and translationally sampled on the binding surface of the receptor together with simultaneously varying the amino acid sequence and side chain conformations. The identity of the two amino acid residues that form the hairpin were held constant in order to maintain the hairpin structure. This process resulted in 200,000 sequences and conformations. These solutions were filtered for interfacial hydrogen bond satisfaction and burial of hydrophobic surface areas. Furthermore, the sequences were screened for their binding energy as predicted by the Rosetta scoring function [38]. The top 30 solutions were visually inspected for hydrogen bond satisfaction, burial of hydrophobic surfaces, and binding surface complementarity, and four favorable candidates by these criteria were chosen for additional rounds of high-resolution docking and sequence design. The resulting peptide sequences were named after the PDB ID containing the original β-hairpin structure. Two out of these four refined structures (pep1jhf and pep1osy) and a peptide directly taken from the EGFR dimerization arm (pepEDA), which is similar to a recently characterized peptide [31], were chosen for experimental characterization (Figure 1B–D).

### 2.2. Synthesis and Cyclization Of Peptides

Peptides pepEDA, pep1jhf, and pep1osy were chemically synthesized for experimental characterization. To assess the effect of conformational constraint, cysteine residues were added at the *N*- and *C*-termini and the peptides were studied in the linear and the cyclic, disulfide-bridged forms (Figure 1E–G). The linear form hereby serves as a control as from design it should not be able to exhibit the same properties compared to the cyclic peptide. Peptides were synthesized using Fmoc/*^t^*Bu-based solid-phase peptide synthesis on Rink amide resin. After cleavage from the resin, linear peptides pep1jhf and pep1osy were oxidatively cyclized under high-dilution in sodium carbonate buffer and ambient atmosphere without any intermolecular cyclization (Appendix A). To avoid methionine oxidation in pepEDA the linear peptide was dissolved in degassed MeOH/water at 1 mg/mL and a solution of I_2_ in MeOH was added to accomplish cyclization. Crude peptides were purified via reversed-phase high-performance liquid chromatography (RP-HPLC). For microscopy studies, the peptides were *N*-terminally labeled with 5(6)-carboxyfluorescein prior to cleavage from the resin (FAM-labeled peptides) and cyclization and purification was performed as described above yielding uniform yellow solids. LC-MS and ESI-HR-MS measurements confirmed the purity and identity of all products (see Appendix A for final compounds S14–S25).

### 2.3. Biophysical Characterization of Peptides

Analysis of the peptides’ secondary structure properties was performed using circular dichroism (CD) spectroscopy. Both the cyclic peptide and the linear precursor were analyzed in comparison (Figure 1E–G and Appendix A). Cyclic pepEDA gave rise to a CD signature that indicated random coil conformation. The linear peptide showed a more flattened curve, typically observed for a flexible disordered structure with a minimum at approximately 195 nm (Figure 1E). No shift of the minimum towards a higher wavelength was observed for pep1jhf cyclic, which suggested no formation of a secondary structure (Figure 1F). A significant β-hairpin character with a maximum at approx. 195 and a minimum at approx. 210 nm was only seen for pep1osy. The linear progenitor in comparison showed an unfolded structure with a minimum at approximately 200 nm (Figure 1G) [39]. Note that the conditions of 10% TFE in water were also chosen for comparison with other, unrelated peptide studies. Despite these mixed results regarding the expected structure, all peptides were further characterized to see if a preformed β-hairpin structure was necessary for biological function. Additionally, all disulfide-bridged peptides were modeled individually in a de novo approach independent of the design method using the peptide structure prediction server PEP-FOLD [40] (Figure 1H–J). The peptide pepEDA (Figure 1H) was modeled as a flexible structure without a significant β-hairpin character. The strong β-hairpin structure detected for pep1osy during CD spectroscopy agreed with the data from structural modeling of pep1jhf (Figure 1J).

Fluorescence polarization assays were performed to investigate the binding (dissociation constant K_d_) of the peptides to the soluble EGFR ectodomain (sEGFR). For pep1jhf and pepEDA, the measurements with lower µM concentrations did not show any binding (Appendix A). Presumably the interaction between peptides and the receptor had higher K_d_ values. Binding assessment of pep1osy was partly successful, yielding an apparent K_d_ of approximately 70 µM (Figure 1B). Data points did not reach a plateau at increasing sEGFR concentrations and thus the maximum anisotropy value was approximated to enable a ballpark data fitting. The concentration of EGFR in these experiments was strongly limited by the solubility of the proteins. Note that the increasing receptor concentrations used in this experiment also drove the competitive dimerization and thus the true peptide affinity was underestimated. The apparent K_d_ value compared favorably to the dissociation constant described for the well-characterized EGFR-binding peptide GE11 (K_d_ = 459 µM) [41].

### 2.4. Cyclic Peptides Show Binding to Cells Expressing EGFR at High and Low Levels

The FAM-labeled peptides were used to assay cell binding properties on cells expressing different levels of EGFR. Experiments were conducted using EGFR-overexpressing A431 cells and low-level expressing MCF7 cells. Both cells lines have been extensively studied regarding their EGFR state [42,43,44]. We first used flow cytometry to assess EGFR binding after incubating cells with peptide at 37 °C for 15 min (Figure 2A and Appendix A). In this assay, the fluorescence signal reported both peptides bound to the cell surface as well as peptides internalized into the cell. Figure 2A shows the normalized mean fluorescence intensity for each sample demonstrating an increase in fluorescence for pepEDA and pep1osy. Interestingly, this increase was also present for the low-level EGFR-expressing MCF7 cell line. The signal on EGFR overexpressing A431 cells was even higher, indicating specific binding of these peptides to EGFR. The fluorescence intensity signal for pep1jhf was comparably low and nearly identical for both cell lines, which may have arisen from to the lack of secondary structure as determined by CD. Further flow cytometry experiments revealed that pepEDA and pep1osy also showed better binding to A431 cells in comparison to pep1jhf in experiments at 4 °C (Appendix A).

Since the internalization of peptides into cells is of special interest when using these peptides as a key for virus-based gene delivery, we analyzed the cellular interaction with confocal laser scanning microscopy. In these experiments, internalization into cells was traced upon the addition of 5 µM fluorescein-labeled peptide for 10 min with subsequent staining of acidic cell compartments using Lysotracker DND-99. Internalization of the EGFR is known to occur via the clathrin-mediated pathway in endosomes which ultimately end up in lysosomes [45,46]. If the FAM-labeled peptide is internalized via endocytosis, it can be identified via colocalization of fluorescence from both the peptide and the lysotracker, taking into account that the FAM fluorescence decreases with pH. Fluorescence images for all cyclic peptides on MCF7 and A321 are presented in Figure 2B. Presence of both fluorescence signals resulted in a white spot in the image overlays that indicated colocalization. White arrows indicate areas where colocalization was detected. Similar to the flow cytometry data, overall staining was low. While peptide binding to the EGFR dimerization interface was expected to inhibit receptor internalization, surface staining of cells was not observed, likely arising from the low surface levels of EGFR in the extended conformation (3–20%) [47] and the low affinity seen in fluorescence polarization. However, internalization was visible for all peptides. Uptake in A431 cells was slightly higher than in the MCF7 cell for pepEDA and pep1osy in agreement with the flow cytometry results. Some co-localization with lysosomes was seen on both cell lines with slightly stronger co-localization in A431 cells, although the low lysosomal pH weakened the FAM-peptide signal [48].

### 2.5. Cyclic Peptide Pep1osy Shows Inhibitory Properties in Wound Healing Assays

Since biophysical measurements are confounded by the intrinsic diversity of receptor conformations and imaging of moderate binding is difficult, we assessed peptide activity using a cell-based functional assay. Epidermoid tumor cell line A431 depends on EGF signaling for proliferation and migration, which can be assessed by wound healing assays monitoring the closure of a scratch in a cell lawn. A431 cells were grown to a confluent cell monolayer, serum-starved, and scratched. The resulting gap was monitored by microscopy and measured over a 6-h period (Figure 3A). As a positive, control cells were treated with 1 nM recombinant human EGF (hEGF) in RPMI medium. Cell migration in µm/h was calculated from a linear regression of the gap area over time plotted against time (Appendix A). For hEGF, a three-fold increase in cell migration compared to that of the negative control (buffer) was observed (Figure 3B). Cyclic peptides were tested either individually to observe their influence on cell migration or simultaneously with 1 nM hEGF to evaluate their antagonistic properties. Incubation of cells with 5 µM peptides did not affect cell appearance, viability, or migration, demonstrating non-toxicity of the peptides towards the cells. Simultaneous incubation of 1 nM hEGF and 5 µM cyclic peptide showed no effect on cell migration for pepEDA and pep1jhf, indicating that these peptides did not sufficiently interrupt EGFR dimerization, consistent with the lack of binding in fluorescence anisotropy experiments. For the detectable binder pep1osy, the cell migration rate decreased to the level of the buffer control, demonstrating a strong inhibitory effect in the presence of hEGF (Figure 3B).

### 2.6. Peptides Grafted on the Recombinant Adeno-Associated Virus 2 Capsid

We grafted each of the three peptides via genetic insertion on the rAAV2 capsid in the exposed 587 loop region, a known permissive site for insertion [49,50]. We hypothesized that the 60-fold multivalent presentation on the approximately 25 nm icosahedral capsid would provide sufficient avidity to overcome moderate receptor affinity and promote virus particle transduction. We speculated that multimeric EGFR-binding and thus clustering might induce viral uptake regardless of EGFR signaling either via EGFR internalization or through increased interaction with secondary AAV receptors.

Naturally, rAAV serotype 2 binds predominately to the primary receptor HSPG. A mutant capsid with the two point mutations, R585A and R588A, which dramatically reduce HSPG binding and lower transduction efficiency, was used as a reference and as a scaffold for grafting. We reasoned that coupling impaired native receptor uptake with designed EGFR binding would result in rAAV with specific targeting towards EGFR overexpressing cells. We further reasoned that the scaffold might stabilize the intended secondary structure of the peptides by paying the entropic cost of fixing the positions of the peptide termini.

The peptides pepEDA, pep1jhf, and pep1osy were genetically incorporated into the capsid corresponding to residue position 587 to retain antiparallel β-sheet hydrogen bond patterns in both the peptide and the scaffold. The resulting viral vectors are named by the peptide (e.g., rAAV2_587_pep1jhf) or for the scaffold (rAAV2_HSPGko). The amino acid sequences of the HSPGko variant and the peptide ligands are given in Figure 4A and a model of the resulting protein in Figure 4B. Viral particles were produced in adherent HEK293 cells by triple-transfection with an ITR plasmid coding for the fluorescent reporter mVenus, a pHelper plasmid, and the peptide-encoding Rep2Cap2 plasmid in ten 100-mm dishes each. Viral particles were purified by ultracentrifugation from a discontinuous iodixanol gradient and the genomic titer was determined with qPCR for each construct. The VP capsid protein ratio of purified viral particles with integrated pep1jhf motif was analyzed using western blot with the anti-VP antibody B1, confirming the presence of capsid proteins VP1, VP2, and VP3 in approximately the expected 1:1:10 ratio (Figure 4C). Final titers ranged from 2 × 10^10^ to 1 × 10^12^ viral genomes per ml (Figure 4D). Only for variant AAV2 587 pepEDA was a slight reduction in vector yield observed, but in general peptide presentation did not strongly interfere with capsid assembly.

### 2.7. Insertion of Designed EGFR-Targeting Peptides Changes the Tropism of rAAV2 towards EGFR Overexpressing Cells

Unlike the analysis of synthetic peptides, no fluorescent labeling was required for viral particles, because upon successful transduction and encapsidation the delivered gene cassette expressed mVenus under the control of a strong viral CMV promoter. We tested transduction of the rAAV2-based viral particles into the fibrosarcoma cell line A431, with a high copy number (approx. > 2 × 10^6^ per cell) of EGFR on its surface and, for comparison, into the breast cancer cell line MCF7 with low expression level of EGFR. Both cell lines were incubated with rAAV2 587 pepEDA, rAAV2 587 pep1jhf, rAAV2 587 pep1osy, and the scaffold control rAAV2 HSPGko at a multiplicity of infection (MOI) of 50,000, and mVenus expression was analyzed 96 h later by flow cytometry (Figure 4E). We repeated these experiments with independent rAAV preparations to corroborate these results (Appendix A). For the negative buffer control and the scaffold rAAV2 HSPGko variant, only negligible transduction of either cell line was observed (Figure 4E). For rAAV2 587 pep1jhf and rAAV2 587 pep1osy, we detected transduction of up to 30% of A431 cells. In contrast, no enhanced cell transduction was observed for either virus on the low EGFR-expressing MCF7 cells. Surprisingly, we did not detect any transduction of A431 cells using rAAV2 587 pepEDA, which could have resulted from the lack of β-hairpin structure determined for this peptide. To get a greater insight into the structural changes happening when inserting peptide sequences into the VP protein, we structurally modeled the peptides in the capsid of AAV2 using SWISS-MODEL (Figure 4F–H). This different analysis method showed that the putative folding of the peptide in the three-fold spike of the capsid does not correlate with the previously generated free peptide models which was to be expected since different constraints were used for calculations.

To get a more complete picture, we also grafted the peptides pep1jhf and pep1osy on different AAV serotype scaffolds. Via genetic manipulation, the peptides were inserted into loops of AAV6 and AAV9 corresponding to the above-described AAV2 modification. However, the determining features of the natural tropisms of these serotypes is less well studied and thus were not knocked down by mutation as done for the AAV2 scaffold. Transduction analyses of A431 and MCF7 cells with each of the constructs resulted in complex but interesting data. Surprisingly, introduction of either peptide did not increase the tropism towards EGFR-overexpressing cells. For AAV9 with low natural transduction in tissue culture, insertions of pep1jhf and pep1osy increased transduction, whereas pepEDA had little influence. For AAV6 with high natural transduction in tissues culture, the peptide insertions strongly reduced transduction (Appendix A).

Taken together, these data suggest a certain degree of specificity of the peptide-modified rAAV2 towards the EGFR presented on A431 cells. At this point it cannot be ruled out that transduction efficiency increased due to other cell-related features or a more complex influence of the insertions on the capsid properties.

### 2.8. rAAV2 587 Pep1jhf Demonstrates Efficacy in Egg Xenografts

Therapeutically more interesting than transduction in tissue culture is the distribution of peptide-presenting AAVs in a whole organism. To analyze this in more detail and yet follow animal welfare reduction, replacement, and refinement guidelines, we tested the transduction of EGFR-expressing tumors xenografts in ovo in chorioallantoic membrane (CAM) assays. CAM assays are frequently used to study tumor growth and vascularization [51,52]. When feasible, CAM assays might provide an initial surrogate for full animal experiments, but to our knowledge this assay type has not yet been tested as tumor model with rAAV. The CAM represents the surrounding of the fertilized chick egg and is composed of a multilayer epithelium. The egg can be opened, and human cancer cells can be transplanted onto the CAM, where they form a vascularized solid tumor. We used this model to study the effectiveness of rAAV delivery to the tumor tissue after systemic injection depending on the capsid modifications with the peptide insertions. Results were compared to PBS control and rAAV2 wt (Figure 5A,B). A431 tumors overexpressing EGFR were grown for five days on the CAM, before PBS or rAAV vectors were injected intravenously. After two days, chicks were sacrificed, and tumors and chick organs were harvested and analyzed by qPCR for their viral DNA content. rAAV2 wt was well-tolerated by the chick and showed in general a low biodistribution in the selected organs which was comparable to experiments performed with mice [53]. Notably, tissue of the kidney showed a higher viral uptake in CAM assays compared to mice (Figure 5B).

To test peptide-mediated EGFR targeting in CAM assays, we focused on rAAV2 587 pep1jhf and rAAV2 587 pep1osy since rAAV2 587 pepEDA did not transduce EGFR overexpressing cells in previous experiments. To justify testing peptide-mediated anti-human EGFR-targeted AAV distribution of rAAV2 pep1jhf and pep1osy in ovo, the structure of the human and the chick EGFR needed to be sufficiently similar at the peptide binding site. Sequence and structure analysis indicated that the proposed binding site of pep1jhf was well-conserved between human and chick (Appendix A). Organ distribution analyses of rAAV2 587 pep1jhf showed higher vector genome copy (VGC) numbers in all analyzed chick organs compared to that of rAAV2 wt, except for the kidney (Figure 5C). rAAV2 pep1osy provided overall only low VGC numbers in all tissues (Figure 5D). This was also reflected when comparing the total VGC summarized from all organs of each rAAV variant (Figure 5E). Injection of rAAV2 587 pep1jhf resulted in severe side effects such as cerebral hemorrhages and gastric bleeding, which were associated with a high embryo mortality. These effects may have arisen from a high number of VGC in both tissues (Figure 5C). Comparison of VGCs in the tumor tissues (Figure 5A–D) revealed that rAAV2 pep1jhf had enhanced tumor targeting properties in comparison to rAAV2 587 pep1osy. The kidney was one of the main targets in these experiments and was thus used for a deeper examination. Comparison of the tumor-to-kidney ratios (Figure 5F) revealed that rAAV2 pep1jhf had enhanced tumor targeting properties in comparison to rAAV2 wt or rAAV2 587 pep1osy.

## 3. Discussion

This study evaluated a de novo approach for peptide design and viral targeting for a specific cell surface marker. From the outset, the project had two biologically demanding design goals: (i) Targeting a protein surface in competition with intra- and intermolecular interactions of the targeted protein (EGFR), and (ii) inhibit a receptor that internalizes upon ligand binding without activating downstream signaling. We tested designed peptides in structurally distinct contexts, comprising free, cyclized, and capsid protein-grafted variants. We used different biological readouts including monomeric cyclic peptides inhibiting wound closure and highly multivalent viral surfaces mediating transduction. The performance of each peptide in the different experiments are summarized in Figure 6. The EGFR-derived peptide pepEDA showed no detectable binding to the receptor, which could be attributed to the low β-hairpin character. Similarly, no binding was observed for the designed peptide pep1jhf, which could also be assigned to the lack of secondary structure detected by CD spectroscopy and corroborated by peptide structural modeling. In contrast, pep1osy showed an affinity towards the EGFR in a fluorescence polarization assay and desired inhibitory effects in the wound healing biological assays. These results were likely based on the β-hairpin character of this peptide.

The sequence of peptide pepEDA is based on published data, which show that this peptide does influence EGFR phosphorylation [29]. In addition, a fluorescein derivative [31], very similar triazolyl-bridged variants [30], and a dimeric form [54] have been studied. Our results were in agreement with published data regarding the CD spectrum of a very similar linear EDA peptide [30] and the observation that a fluorescently-labeled version was only detected within the cell but not on the surface by laser scanning microscopy [31,55]. In our wound healing assay, pepEDA produced no effect, which agreed with the previously detected low influence of similar peptides on EGFR signaling. In this light, the computationally-designed pep1osy showed a much better performance in cellular assays and, thus, promising results for further experiments [31]. We believe that after this smaller set of experiments, a deeper analysis of the binding mode as well the influence on the intracellular signaling and EGFR recycling is an attractive next step.

Behavior of the designed peptide sequences can vary greatly when incorporated into rAAV capsids when compared to free peptides. These changes likely arise from local structural effects including specific conformational rigidification of the peptide termini and influence of the surrounding protein environment. mVenus-expressing rAAV vector particles that carry the peptides pep1jhf or pep1osy in surface-exposed capsid loops showed strong transduction of EGFR-expressing cell lines in vitro. In contrast, the rAAV variant presenting pepEDA on the surface showed no transduction ability. A possible reason could be failure to form the β-hairpin during protein assembly that could impair interaction with the receptor. Interestingly, rAAV-bearing peptide pep1jhf showed strong transduction despite having undetectable binding by fluorescence polarization. We hypothesized that the β-hairpin conformation of the peptide was stabilized by the local structure of the 587-loop of the VP protein, yielding the notable possibility that peptides showing undesirable biophysical properties in their free form may have attractive features for gene therapy in the context of viral capsids. At this point, we were inclined to assume that only the peptide mediated the tropism of the AAV. When repeating the experiments with different AAV scaffolds (AAV6 and AAV9), we observed strong effects but not an enhanced specificity towards the EGFR. We concluded that the interplay between the peptide, the knock down of the natural tropism, and the scaffold do strongly influence the targeting of the AAV. This interplay includes possible changes of capsid properties for DNA loading during production as well as particle trafficking from the cell surface to the nucleus and final DNA release during transduction. Such observations were in line with previously published data from Börner et al. [56].

Experiments in chicken embryo xenografts showed that the biodistribution patterns of rAAV2 wt nearly completely mirrored those observed in mice [53]. The main difference was seen in the comparably high VGC numbers in the kidney tissue. Literature suggests that in chickens, the kidney is the primary secretory organ [57]. It is therefore likely that the high DNA copy number of AAV2 in the kidneys is due to a virus clearance effect. We observed a great divergence between cell culture and in ovo experiments between rAAV2 587 pep1osy and rAAV2 587 pep1jhf. Much higher viral uptake in chick organs and human A431 tumors grown on the CAM was detected for rAAV2 587 pep1jhf. The increased viral uptake in the tumor does correlate with the observation of less VGC detected in the kidneys, as a successful transduction of the target tissue would lead to a decrease in clearance via the kidneys. The difference of AAV2 587 pep1osy to cell culture experiments might be explained by enhanced blood clearance and poor vascular escape. A further explanation for the low copy numbers from the chicken organs might be related to the structural differences between the human and the chicken EGFR, even if sequence and structure comparisons showed conservation across the binding site. Infectivity of vector preparations used for in ovo experiments was tested in cell culture experiments prior to systemic administration. In contrast, rAAV2 587 pep1jhf not only showed the great transduction abilities of the tumor tissue, but also had elevated levels in other tissues of the chicken.

As a growth factor receptor, EGFR is likely to be widely overexpressed in fetal tissues and thus viral uptake into all organs is probable. The high lethality after systemic injection due to severe cerebral hemorrhages and gastric bleeding might be explained by the high VGCs found in these organs. Literature suggests that EGFR expression increases in the stomach during embryogenesis, which might further explain these side effects [58]. The elevated levels in the brain are surprising as the rAAV2 wt variant was not detected and it is known that rAAV2 is not able to cross the blood–brain barrier [59]. However, it was shown that the blood–brain barrier of one-day-old chickens is permeable, which could allow the retargeted rAAV vector to enter the brain either through tight junctions between the endothelial cells or through transcytosis via the endothelial cells [60]. EGFR expression in the fetal brain is elevated, which might contribute to a higher viral uptake.

Overall, the in ovo experiments confirmed the distribution that was previously described for rAAV wild-type vectors in mouse experiments with the exception of a higher VGC detected in the kidney in CAM assays [53]. These findings thus provide information to help avoid more time-consuming experiments in mice. Experiments conducted for the rAAV variants show that the transfer from an in vitro cell culture-based system to an in vivo model can be complex. Here, the transfer from cell culture to in ovo experiments was successful for rAAV2 587 pep1jhf but not for rAAV2 587 pep1osy, demonstrating the importance and utility of the in ovo experiments in a therapeutic development pipeline. The severe complications observed after viral vector injection showed the need for further optimization of the viral vector. To this end, a precise analysis of the brain regions could be of interest to investigate if side effects correspond to reported EGFR-expression levels.

## 4. Materials and Methods

### 4.1. Structural Modeling

Structures of cyclized chemically synthesized peptides were predicted using PEP-FOLD 2.0 (https://bioserv.rpbs.univ-paris-diderot.fr/services/PEP-FOLD/) [40,61]. Resulting clusters were ranked according to their lowest energy (score optimized potential for efficient structure prediction (sOPEP)), and the five best 3D models for each peptide were selected. The resulting structural coordinates were visualized using the PyMOL program (DeLano, W. L. (2002); The PyMOL Molecular Graphics System 2.0, Schrödinger, New York, NY, USA).

Modeling of peptides incorporated into the VP protein was accomplished using the online tool SWISS-MODEL (https://swissmodel.expasy.org/) [62]. Peptide sequences were incorporated into the VP sequences of AAV2 at position 587. The PDB coordinates of the wild-type AAV were used as a template to generate the homology model for the peptide-modified sequence. Models were calculated for the monomer and the homo-60-mer. Results were visualized using the PyMOL program.

### 4.2. General Procedure

Fmoc-amino acids, Cys(Trt), Tyr(*^t^*Bu), Asn(Trt), Pro, Thr(*^t^*Bu), Gln(Trt), Met, Glu(*^t^*Bu), Ser(*^t^*Bu), Lys(Boc), Ala, Gly, Trp, Val, Phe, Arg(Pbf), and Leu were obtained from Iris Biotech GmbH (Marktredwitz, Germany) and Carbolution Chemicals GmbH (St. Ingbert, Germany). 

Analytical RP-HPLC was performed on a Shimadzu Nexera XR UHPLC equipped with a pump (LC-20AD), a column oven (CTO-2CA), a diode array detector (SPD-M20A), an autosampler (SIL-20AXR), and a communication module (CBM-20A) using a Phenomenex Luna C18 column (3.0 µm, 100 × 2.0 mm). Analytical LC-MS (Phenomenex Luna C18 column, 3.0 µm, 100 × 2.0 mm) and high-resolution mass spectroscopy (HRMS) (Phenomenex Luna C18 column, 3.0 µm, 100 × 2.0 mm) was executed on a time-of-flight mass spectrometer (Agilent 6220 TOF-MS) with a dual electrospray ionization (ESI) source, and an HPLC system (HPLC 1200) with an autosampler, binary pump, column oven, degasser, and a diode array detector operating with a spray voltage of 2.5 kV. A nitrogen generator (NGM11) produced nitrogen that served both as nebulizer and dry gas. The ESI-L tuning mix was used for external calibration.

MALDI-TOF-MS analyses were conducted using an Ultraflex (Bruker, Bremen, Germany) equipped with a 355 nm Nd:YAG laser, 50 Hz in positive mode at 1000 shots/spectrum using 2,5-dihydroxybenzoic acid (DHB), or α-cyano-4-hydroxycinnamic acid (CHCA) as the matrix. Calibration was performed with polyethylene glycol (PEG) 400–1200. Preparative HPLC on a Merck–Hitachi LaChrom HPLC (interface D-7000, pump L-7150, detector L-7420) was performed using a Hypersil Gold C18 column (1.9 µm, 250 × 21.2 mm) or a Hypersil Gold C18 column (7 µm, 250 × 10.0 mm).

### 4.3. Peptide Synthesis

Peptides were synthesized using the Fmoc/tBu-strategy according to standard protocols in a plastic syringe fitted with a polypropylene porous disk at room temperature. Each reaction was followed by washing five times with 10 mL DMF per 1 g resin. Solvents and volatile reagents were removed by partial vacuum. Coupling steps were performed on a horizontal shaker and reactions controlled by Kaiser test and/or MALDI or LC-MS. Steps involving 5(6)-carboxyfluorescein were carried out protected from light.

Rink-amide resin (loaded with 0.5 mmol/g) was swollen in DMF (10 mL/g) for 15 min and deprotected twice with 20% piperidine/DMF, 0.1 M 1-hydroxybenzotriazole (HOBt) for 20 min, and then washed. Fmoc-Cys(Trt)-OH (4 equiv) and ethyl cyanohydroxyiminoacetate (Oxyma) (4 equiv) were dissolved in DMF, combined, *N*,*N*′-diisopropylcarbodiimide (DIC) (4 equiv) was added, and the mixture was shaken for 30 s. This reaction mixture was added to the resin and incubated for 2 h at room temperature. Next, the resin was filtered, washed, and capping was carried out twice with acetic anhydride (10 equiv) and pyridine (10 equiv) in DMF. The resin was dried, and loading determined (0.44 mmol/g). Each coupling step was repeated twice. Fmoc-Xaa-OH (4 equiv) and Oxyma (4 equiv) were dissolved in DMF and 4 equiv DIC were added. The solution was inverted for 30 s and afterwards incubated with the resin for 1 h.

### 4.4. N-Terminal Fluorescein Labeling

5(6)-carboxyfluorescein (3 equiv) was coupled to the resin using Oxyma (3 equiv) and DIC (4 equiv) overnight in the dark. Residual 5(6)-carboxyfluorescein was washed from the resin twice with a solution of 20% piperidine/DMF for 20 min. For pep1osy, coupling was incomplete. In a second attempt, 12 equiv of 5(6)-carboxyfluorescein and PyAOP were incubated in a mixture of DMF and *N*-methyl-2-pyrrolidone (NMP) in the dark for 72 h [63]. Again, two incubation steps with 20% piperidine/DMF were performed before monitoring the reaction using MALDI.

### 4.5. Peptide Cleavage

In a final step, peptides were cleaved from the resin with a mixture of TFA/H_2_O/TIPS/DTT (92.5:2.5:2.5:2.5 *v*:*v:v:w*) for 2 × 3 h each. Due to the presence of methionine, the cleavage solution of pepEDA was purged with argon. The solvent was evaporated under reduced pressure and peptides pep1jhf and pep1osy were precipitated in ice cold ether followed by a freeze-drying step. Peptides to be cyclized were used in the next step without further purification, while the linear peptides were purified by RP-HPLC.

### 4.6. Cyclization of Peptides

Cyclization of linear peptides pep1osy and pep1jhf was performed at a dilution of 1 mg/mL in 0.1 M NaHCO_3_ buffer (pH 7.8). The reaction was stirred open to atmosphere until completion (2–24 h). Cyclic peptides were desalted on a manual-operated C18 column (25 g, 400–220 mesh) using water/0.1% TFA (200 mL) followed by elution with MeOH/0.1% TFA (300 mL). The solvent was evaporated under reduced pressure and the crude peptides were freeze dried before purification using RP-HPLC. Cyclization of pepEDA required special conditions due to the methionine residue. The linear peptide was dissolved in degassed MeOH/water (5:95) at 1 mg/mL. I_2_ diluted in MeOH (0.06 M) was added dropwise to the diluted peptide until the solution remained slightly yellow (approx. 3 mL) [64]. Upon completion (0.5–1 h) the reaction was quenched with 1 M ascorbic acid resulting in a colorless solution. FAM-labeled peptides were handled protected from light. Analytical data for the final peptides are given in Appendix A.

### 4.7. Cell Culture

HEK293 cells (ACC 635, DSMZ, Braunschweig, Germany) were cultured in Dulbecco’s modified eagle medium (DMEM). A431 and MCF7 cells (ACC 91 and ACC 115, DSMZ) were cultured in RPMI and frequently negatively analyzed for mycoplasma. Both media were supplemented with 10% (*v*/*v*) fetal calf serum (FCS) and 1% (*v*/*v*) penicillin/streptomycin (P/S) (Sigma Aldrich, Schnelldorf, Germany and incubated at 37 °C and 5% CO_2_. For medium not already supplemented with glutamine, 1% penicillin/streptomycin/glutamine (Sigma Aldrich) was added prior to use.

### 4.8. Wound Healing Assay

A431 cells (ACC91, DSMZ) were seeded in 24-well plates at a density of 2 × 10^5^ cells/well (TC-treated, Sarstedt) in RPMI-1640 culture media (D8758, Sigma Aldrich; supplemented with 10% FCS and 1% P/S (P4333, Sigma Aldrich)) and incubated at 37 °C and 5% CO_2_ for 24 h until a confluent monolayer was formed. A serum-starving step was performed overnight in RPMI-1640 without FCS. Per well, two scratches were made using a sterile 10-µL pipet tip. Dislodged cells were washed off the plate with PBS thrice and remaining cells treated with the test substance at a specified concentration (e.g., 1 nm EGF) in RPMI-1640 with 1% P/S. Control cells received an equal volume of buffer (1 mL RPMI-1640 with PBS). Cells were maintained at standard conditions and cell migration was observed using a Leica DMI6000B microscope at selected time points ranging from 0 to 5 h of incubation. Identical positions were analyzed by the “Mark and Find” feature of Leica control software in combination with a high-precision motorized stage. The MRI wound healing tool plug-in (Volker Bäcker) for ImageJ version 1.51 was applied to determine the area of cell-free surfaces [65]. Linear regressions of the surface area versus time were used to determine the cell migration rate in Origin2019 (OriginLab) using $v_{migration}=\frac{\left|Slope\right|}{2\;\times \;length\;of\;the\;gap}$. Only the first data points that showed a linear correlation were included in the regression. The length of the gap was measured using ImageJ. Data points from two independent experiments were analyzed.

### 4.9. Confocal Fluorescence Microscopy

Live cell microscopy was performed on the inverted confocal laser scanning instrument Zeiss LSM780 (Zeiss, Oberkochen, Germany) to visualize binding and internalization of FAM-labeled peptides. Cells were seeded at a density of 1.5 × 10^4^ per well in an 8-well µ-slide (Nunc, LabTek, Thermo Fisher, Karlsruhe, Germany) in 300 µL RPMI medium and incubated overnight. Microscopy was performed while maintaining cell culture conditions in an incubation chamber at 37 °C. Cells were incubated with 10 µL NucBlue Live ReadyProbes Reagent (Invitrogen) and 50 nm Lysotracker DND-99 (Invitrogen) for 10 min. Peptides were diluted to 100 µM stock solutions in DMSO and added to the cells with 5 µM concentration for 10 min at 37 °C (final DMSO concentration 5%). Residual peptide was removed with five extensive washing steps with RPMI (*w*/*o* phenolred and FCS) before imaging the cells with a 63× objective (LCI Plan_neofluar 63×/1.3 Imm Korr DIC M27) using immersion oil (Immersol W (2010), Zeiss). Laser and detector ranges were used with corresponding main beam splitters as given in Table 1. Image acquisition was performed using Zeiss Zen 2011. Image analysis was carried out using ImageJ (National Institute of Mental Health, Bethesda, MD, USA). A macro was used to analyze areas of colocalization between the carboxyfluorescein signal of the FAM-labeled peptide and the Lysotracker DND-99 signal.

### 4.10. Circular Dichroism Spectrometry

Circular dichroism spectra were recorded with a J-810 spectropolarimeter (Jasco, Pfungstadt, Germany) equipped with a Peltier-type temperature controller. Peptides were measured at a concentration of 100 µm in 10% TFE/water (*v*/*v*). Each spectrum was recorded thrice from 190 to 250 nm at 25 °C with a scanning rate of 50 nm/min. Data was analyzed using Origin2019.

### 4.11. Fluorescence Polarization

Fluorescence polarization assays can be utilized to evaluate binding affinities of FAM-labeled peptides towards larger proteins like the soluble EGFR (sEGFR). The sEGFR was serially diluted in 0.05% Tween^®^20/PBS (FP buffer) starting with a concentration of 25 µM (2.5× dilution). Then, 15 µL of the dilutions were pipetted into 384-well plates (BRANDplates^®^, 384-well, pureGrade™, black). The FAM-labeled peptide stock (20 µM) was diluted to a working stock concentration of 40 nM in FP buffer. A volume of 5 µL peptide working stock (10 nM final) was added to each well. Each sample was measured in technical triplicates. The plate was incubated at 4 °C for 1 h before centrifugation at 1000× *g*, 2 min. Fluorescence polarization data was acquired using a Tecan Spark 10M at Ex 485/20 nm and Em 520/20 nm. Fluorescence polarization was converted into anisotropy and analysis of experimental data was performed in Origin2019. The 1:1 binding model equation is shown below with r as anisotropy, $r_{0}$ as anisotropy of the free peptide, $r_{b}$ as anisotropy of the EGFR:FAM-labeled peptide complex, $K_{d}$ as dissociation constant, ${\left[L\right]}_{t}$ as total labeled ligand concentration, and ${\left[P\right]}_{t}$ as total protein concentration [66].
$$r=r_{0}+\left(r_{b}-r_{0}\right)\frac{K_{d}+{\left[L\right]}_{t}+{\left[P\right]}_{t}-\sqrt{{\left(K_{d}+{\left[L\right]}_{t}+{\left[P\right]}_{t}\right)}^{2}-4{\left[L\right]}_{t}{\left[P\right]}_{t}}}{2{\left[L\right]}_{t}}$$

### 4.12. Viral Particle Production

HEK293 cells were seeded in 100-mm dishes at a density of 3 × 10^6^ cells in DMEM the day before transfection. A total amount of 15 µg DNA per dish was transfected using calcium phosphate. A triple transfection of the Rep2Cap plasmid, an ITR2-containing plasmid, and the pHelper plasmid (1:1:1 molar ratio) was performed as described previously [67]. Cell were incubated at 37 °C for 72 h, harvested, and separated from the medium by centrifugation (2000× *g*, 5 min).

### 4.13. Purification of Viral Particles

Harvested cells were resuspended in lysis buffer (50 mM Tris, 150 mM NaCl, 2mM MgCl_2_, pH 7.5). Three freeze-thaw cycles alternating between a dry ice-ethanol bath and a 37 °C water bath released viral particles from cells. DNA contamination was degraded by incubation with Benzonase (final 100 U/mL, Sigma Aldrich) at 37 °C for 1 h prior to addition of CHAPS (*w*/*v*) at 0.5% final). Crude lysate was cleared from cell debris by centrifugation (3000× *g*, 10 min) and further purification was performed with a discontinuous iodixanol gradient (Progen Biotechnik, Heidelberg, Germany) as described [68]. The lysate was transferred onto a gradient of 60%, 40%, 25%, and 15% iodixanol) using a syringe equipped with a 23G (0.60 × 80 mm) injection needle in an open top polyallomer 16 × 76 mm tube (Science Services. Tubes were sealed and centrifuged in a T-880 rotor (Sorvall, Thermo Fisher Scientific, Langenselbold, Germany) at 340,000× *g* for 2 h at 18 °C. A 21G (0.80 × 40 mm) injection needle was used to collect the AAV-containing fraction from the bottom of the tube. Iodixanol was removed and buffer was exchanged to 1× HBSS (Sigma Aldrich) via Amicon Ultra-4 100K centrifugal filter units (Merck Millipore, Darmstadt, Germany).

### 4.14. SDS-PAGE and Western Blot Analysis

Samples were incubated with SDS loading buffer at 95 °C for 10 min followed by centrifugation for 1 min at 10,000× *g*. Then 20 µL per lane were loaded on a 10% SDS-polyacrylamide gel (Hoefer SE260 system, Serva, Heidelberg, Germany). Blotting was carried out onto a 0.45 µm nitrocellulose membrane (Thermo Fisher Scientific, Schwerte, Germany) using a semi-dry blotting system (V20-SDB, Scie-Plas, Cambridge, UK) following standard protocols [69]. The membrane was blocked with 10% (*w*/*v*) non-fat milk in TBS before incubation with the VP-specific primary antibody B1 (1:100, anti-AAV VP1/VP2/VP3 mouse monoclonal supernatant (Progen Biotechnik, Heidelberg, Germany)) in blocking buffer for 1.5 h. Detection was performed after incubation with a secondary anti-mouse IgG linked to a horseradish peroxidase (1:5000, anti-mouse IgG, HRP-linked antibody, Cell Signaling Technology, Leiden, The Netherlands) by luminescence detection (Pierce ECL western blot substrate, Thermo Fisher Scientific).

### 4.15. Determination of Genomic Titers

Next, 5 µL viral sample were incubated with 10 U DNase I (New England Biolabs) and 10× DNase I buffer to a total volume of 50 µL at 37 °C for 30 min followed by a 20 min heat inactivation at 75 °C. A fivefold dilution of the DNase I digest was prepared and 5 µL were mixed with 2.5 µL primer qPCR-hGH-for (5’-CTC CCC AGT GCC TCT CCT-3’) and 2.5 µL qPCR-hGH-rev (5’-ACT TGC CCC T TG CTC CAT AC-3’) at a final concentration of 4 µM and 10 µL of 2× GoTaq qPCR Mastermix (Promega, Walldorf, Germany) to a final volume of 20 µL. The qPCR reaction was carried out following the manufacturer’s instructions (Promega) using a LightCycler 480 II (Roche, Mannheim, Germany). The genomic titer was calculated from a standard curve of 10 to 10^6^ copies of the ITR plasmid (pZMB0522 containing the hGH polyA motif) with an efficiency between 90–110% and an R value less than 0.1.

### 4.16. Transduction Assay

Transduction assays were performed on 12-well plates with 10,000 cells seeded per well. Viral samples were applied to each well and incubated for 96 h. Cells were prepared for flow cytometry by trypsinization, and samples were resuspended in PBS. Ten thousand events were counted on a FACSCalibur (Becton–Dickinson, Heidelberg, Germany). All transduction assays were performed in biological duplicates in two independent experiments with rAAV preparations from two different batches. Data analysis was performed using FlowJo V10 and Origin2019.

### 4.17. Chorioallantoic Membrane (CAM) Assay

Fertilized eggs at embryonic day 0 were incubated in horizontal position at 37.8 °C and 58–60% relative humidity (ProCon Grumbach, Compact S84 with automatic turning trays) after being carefully cleaned with tap water. Vibration was prevented by the automatic turning of the egg five times by 180° at 6 °/s. After 96 h, a small hole was punctured into the pointed end of the egg. The eggs were incubated for 10 min in an upright position to allow air to escape and deepening of the egg content. The hole was sealed with a patch (Leukosilk S) before enlargement of the hole with scissors to a diameter of 1 cm. After closing the larger hole with a second patch, eggs were incubated in an upright position without turning until day 7, where a 1-mm thick silicon ring with an inner diameter of 5 mm was placed on the CAM atop a branching vessel. To prevent dry out, the opening of the egg was closed with a patch during all incubation steps. On the next day, A431 cells were prepared by detaching from adherent culture using Accutase^®^ before washing with PBS and resuspension in serum-free medium. The suspension of 3 × 10^6^ A431 cells in a final volume of 15 µL was mixed with 10 µL ice-cold Matrigel^®^ (Corning) and added to the silicon ring on the CAM. After 12 days, a second enlargement of the hole was conducted to allow for intravenous injection of 1 × 10^11^ rAAVs in 100 µL PBS with a 27G (0.40 × 20 mm) cannula. The cannula was left at the site of injection to prevent severe bleedings due to poor coagulation of the chick blood. After removing the syringe, the cannula was sealed with a silicon plug and fixed with several patches. At day 14, chick embryos were sacrificed by systemic injection of 3 mg propofol per egg before extraction of tumors and organs. All biologic samples were deep-frozen in liquid nitrogen and stored at −80 °C. DNA isolation from organs was performed using the GenElute™ Mammalian Genomic DNA Miniprep Kit. Organs were thawed and small pieces of organs were shredded in 180 µL lysis buffer T. Proteinase K (20 µL) was added, and the samples incubated at 56 °C overnight to assure complete digestion of the tissue. Further steps were performed in accordance with the manufacturer’s instructions. Finally, DNA was eluted in 200 µL Tris (10 mM, pH 8.0) and stored either at 4 (short term) or −80 °C (long term). Viral DNA was quantified from samples of isolated DNA. The DNA concentration was determined photometrical and samples were diluted in ddH_2_O to a final concentration of 10 ng/µL. The qPCR reaction was performed using 10 µL Kapa SYBR FAST (# KK4602 Kapa Biosystems, Roche, Mannheim, Germany), 0.4 µL primer forward (concentration of 10 pmol/µL), 0.4 µL primer reverse, and 7.2 µL ddH_2_O with 2 µL DNA sample per well (20 ng DNA total). Sequences of primer pairs are given in Table 2. The PCR program was: (i) 10 min 95 °C; (ii) 40 cycles of 30 s 95 °C, 30 s 60 °C, 20 s 72 °C; (iii) 10 min 72 °C; (iv) denaturation cycle to confirm homogeneity of amplified DNA. The viral copy number was additionally normalized to the chick or human (for tumor tissues) actin copy number by multiplying the C_t_ value of each sample with the quotient of C_t_ (actin average)/C_t_ (actin sample). A standard curve from 1 × 10^9^ to 1 × 10^2^ viral genomes per well was used to finally quantify the viral DNA amount.

## Figures and Tables

**Figure 1 ijms-21-09535-f001:**
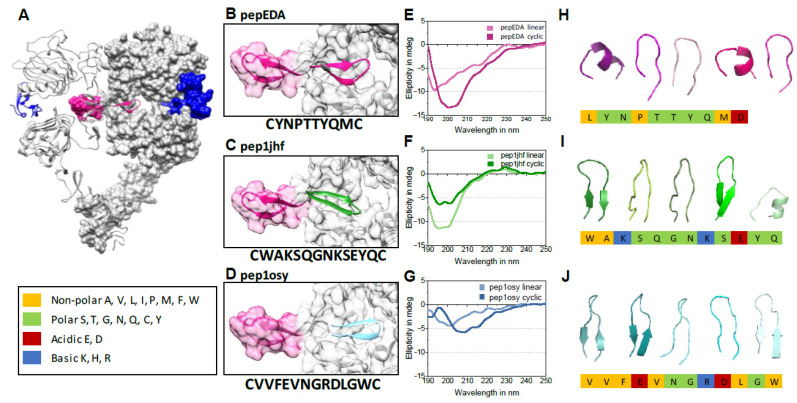
Epidermal growth factor receptor (EGFR) crystal structure, peptide binding models, and circular dichroism (CD) spectra. (**A**) Cocrystallization of the dimerized human EGFR extracellular domains and the natural ligand EGF (blue) (protein databank (PDB) 3NJP). The EGFR dimerization arm (EDA) is highlighted in magenta in both domains. (**B**–**D**) Model structures of designed peptides docked into the dimerization interface. Ribbon representations of peptides are highlighted in color, peptide names are inset, and peptide amino acid sequences are listed below the image. (**E**–**G**) Circular dichroism (CD) spectra of linear and cyclic peptides (100 µM, 10% TFE/water (*v*/*v*), 190 to 250 nm, 25 °C) (for absorption spectra see Appendix A). (**H**–**J**) 3D structure models of disulfide-bridged peptides as calculated by the online tool PEP-FOLD 2.0 and the top 5 clusters are shown in ribbon structure as created using the program PyMOL. The chemistry of the individual amino acid residues is color-coded according to the legend on the left.

**Figure 2 ijms-21-09535-f002:**
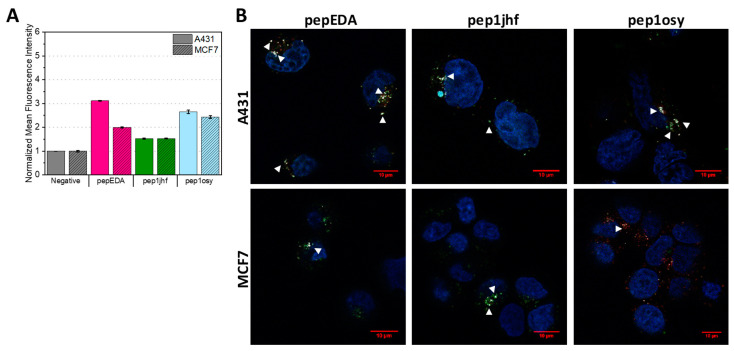
Binding properties of EGFR-targeting peptides to A431 and MCF7 cells. (**A**) Flow cytometry analysis of cyclic peptides incubated with A431 (uniform color) and MCF7 (shaded color) cells at 20 µM for 15 min at 37 °C. Data are represented as mean ± SD for two independent experiments and are given relative to the negative (i.e., buffer control). Raw data are given in Appendix A. (**B**) Live cell imaging of cyclic FAM-labeled peptide variants incubated with A431 and MCF7 cells for 10 min at 5 µM. Nuclei and lysosomes were counterstained using NucBlue and Lysotracker DND-99, respectively and merged images are shown. Colocalization of Lysotracker DND-99 and FAM-labeled peptide is indicated with white triangles. Fluorescence microscopy was performed at 63× magnification using an inverted laser scanning Zeiss LSM780 microscope. Scale bars are highlighted in red and represent 10 µm. Single images can be found in Appendix A.

**Figure 3 ijms-21-09535-f003:**
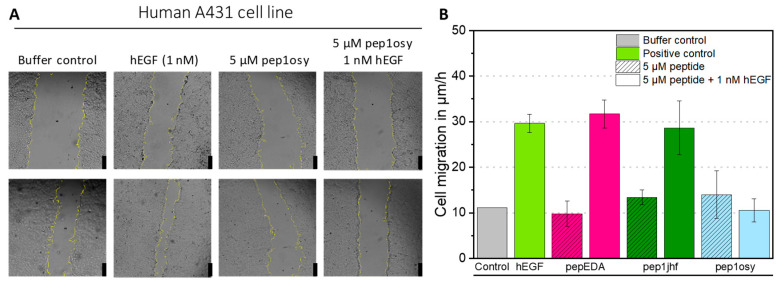
Effect of EGFR-directed peptides in the presence of the natural EGFR-ligand EGF in a wound healing assay with A431 cells. (**A**) A431 cells were incubated with a premixed solution of 1 nM hEGF and 5 µM peptide (here as example pep1osy) or with a 5 µM peptide solution only, and bright field images were taken with an automated inverted microscope (Leica) over 6 h. Here, images after 0 h (upper row) and 6 h (bottom row) are given as examples. The yellow cell borders were detected by image recognition using ImageJ. Scale bars given as black bars represent 250 µm. (**B**) Bar chart presenting the cell migration velocity as determined from microscopy images. Calculations are based on linear regressions shown in Appendix A. The migration is given for each peptide (shaded color) and each peptide mixed with 1 nM hEGF (uniform color) Experiments were performed in biological duplicates and data is represented as mean ± SD.

**Figure 4 ijms-21-09535-f004:**
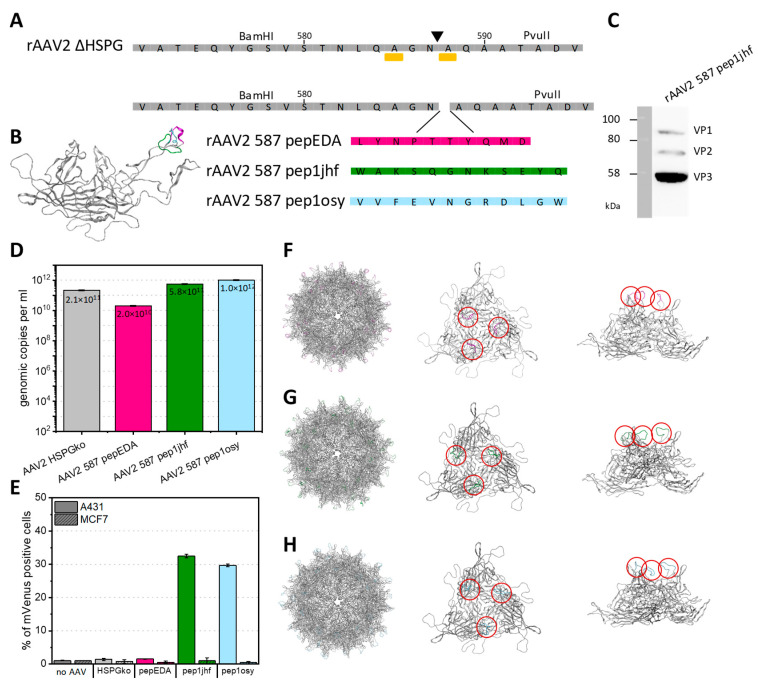
Insertion of peptides at residue 587 into AAV2 VP2 and production and transduction. (**A**) VP protein as illustrated by Chimera from PDB 1LP3 and amino acid sequences of VP with mutations R585A and R588A (yellow sidechain, bar under sequence) and corresponding restriction sites BamHI and PvuII enabling respective genetic modifications. (**B**) Superimposition of VP3 monomers of AAV2 showing all peptide modifications in the 587 position. (**C**) Western blot analysis of recombinant adeno-associated virus (rAAV2) 587 pep1jhf showing the presence of all three VP proteins after incubation with the anti-VP antibody B1 (Progen). Full raw images are given in Appendix A. (**D**) Effects of peptide integration on production yield. rAAVs were produced in a triple-transfection HEK293 system. Viral particles were purified using iodixanol gradients and DNaseI-resistant genomic copies were quantified via qPCR against plasmid standards and are presented as genomic copies per ml. Error bars represent mean ± SD for two replicates. (**E**) Quantification of mVenus-positive A431 or MCF7 cells transduced with mVenus-gene carrying rAAV presenting the peptide indicated in the category axis at a multiplicity of infection (MOI) of 50,000. mVenus expression levels of biological duplicates were analyzed 96 h post transduction by flow cytometry. Error bars represent mean ± SD for two replicates. Pep1osy- and pep1jhf-carrying AAV2 show an enhanced transduction of EGFR-expressing cells. (**F**–**H**) Capsid structure, side-view, and view along the threefold symmetry axis of the trimer for pepEDA (**F**), pep1jhf (**G**), and pep1osy (**H**). Structural calculations were performed using the SWISS-MODEL online tool (https://swissmodel.expasy.org/) and were visualized using PyMol. Red circles highlight the positions of the peptides within the trimer.

**Figure 5 ijms-21-09535-f005:**
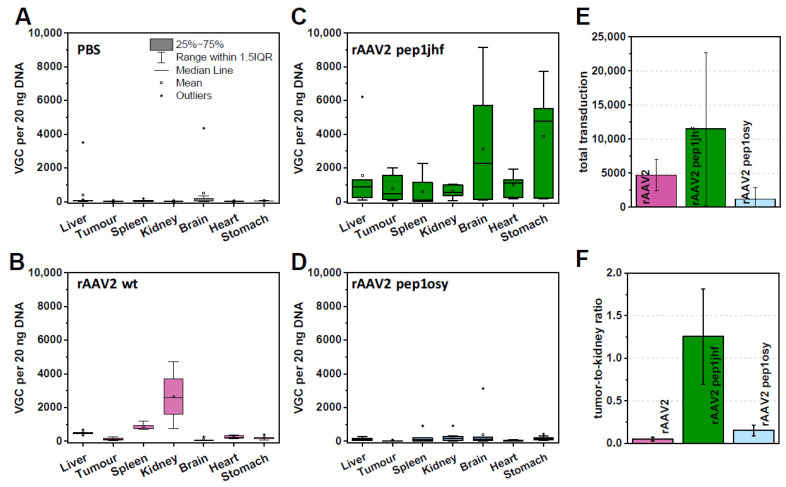
Chorioallantoic membrane (CAM) assays analyzing rAAV variants with respect to viral uptake in organs and tumor xenografts. A431 tumors were grown on CAM for five days prior to injection of rAAV2 variants with 1 × 10^11^ viral particle per egg. After 48 h incubation organs were extracted from chicken embryos, DNA was isolated, and vector genome copy (VGC) of 20 ng total DNA quantified by qPCR analyses. Box-whisker diagrams represent indicated replicates for each sample: (**A**) PBS, 10 replicates; (**B**) rAAV2 wt, 5 replicates; (**C**) rAAV2 pep1jhf, 7 replicates; (**D**) rAAV2 pep1osy, 11 replicates. (**E**) Total VGC uptake of each rAAV variant was calculated by summarizing the VGC per 20 ng DNA for each organ. (**F**) Tumor–kidney ratio was calculated from mean copy number/20 ng DNA for each organ. Error bars represent mean ± SEM for replicates.

**Figure 6 ijms-21-09535-f006:**
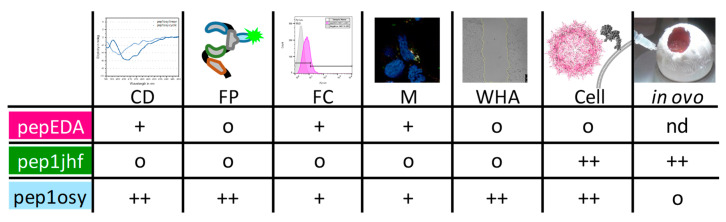
Summary of peptide properties determined using circular dichroism (CD), fluorescence polarization (FP), flow cytometry (FC), microscopy (M), wound healing assay (WHA), and transduction experiments in cell culture (Cell) and in CAM assays (in ovo). Peptides were ranked regarding their performance in each experiment from o (neutral/negative), + (good), and ++ (very good).

**Table 1 ijms-21-09535-t001:** Laser and detector ranges used during live cell imaging.

Channel	Color	Laser line	Detector Range
NucBlue (Hoechst 33342)	Blue	405 nm	414–417 nm (Ch1)
5(6)-Carboxyfluorescein	Green	488 nm (argon)	499–553 nm (ChS1)
Lysotracker-DND99	Red	594 nm	609–700 nm (Ch2)

**Table 2 ijms-21-09535-t002:** Oligonucleotide primers used for viral DNA quantification in qPCR reaction.

Name	Sequence 5′-3′
βActin (gallus gallus) for	ATTGCCCCACCTGAGCGCAA
β-Actin (gallus gallus) rev	CATCTGCTGGAAGGTGGACA
β-Actin (human) for	GCTCCTCCTGAGCGCAAG
β-Actin (human) rev	CATCTGCTGGAAGGTGGACA
mVenus for	AAACTGATTTGCACCACCGG
mVenus rev	GCAAAGCATTGCAGGCCATA

The β-actin primer pair (*Gallus gallus*) was used for organs, while the β-actin primer pair (human) was used for the tumor tissue. β-actin served as a house-keeping gene for normalization of the data set.

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
