# Peer review of "EGFR-Binding Peptides: From Computational Design towards Tumor-Targeting of Adeno-Associated Virus Capsids"

_ijms, 2020, doi:10.3390/ijms21249535_

Round 1

Reviewer 1 Report

The authors present a robust methodology based on the combination of a computational design approach with rational capsid modification to target viral vectors. The methodology is well described and the synergy shown by the dual theoretical/experimental is excellent. Very good paper, which holds promise for future developments in gene therapy.

Author Response

We thank the Reviewer for this positive assessment.

Reviewer 2 Report

In their paper, Feiner and colleagues describe the computational design and experimental characterisation of EGFR-binding peptides. The study is motivated by the fact, that the successful de novo design of protein, and in particular receptor-binding peptides might significantly reduce the time and effort spent on screening large numbers of random peptides using traditional approaches such as peptide display methods.

I certainly appreciate the effort the authors put into characterising their lead candidates using a variety of biophysical, in vitro cell culture and in vivo assays. Most notably, peptides were not only tested in their monomeric form but were further incorporated into the structural VP proteins of recombinant adeno-associated viruses to compensate for the relative low affinities by enhancing avidity.

Even though the results of different sets of experiments seem to be hard to correlate, the authors do discuss this in the final part of the manuscript without overemphasising the significance of their results. This study is definitely interesting for colleagues working on similar projects and I do recommend the manuscript for publication. However, I have several (partly major) comments that need to be addressed in either writing or additional experiments.

Major:

  1. It’s relatively hard to judge whether the apparent Kd for pep1osy can be derived from the fluorescence polarization assays. In my opinion, there are not enough datapoints to fit this curve reliably. The authors should consider using a complementary technique such as SPR or ITC to test the affinity of the peptides (characterisation of GE11 as described in ref 39 might be helpful in this respect). Alternatively, a phage ELISA with phage particles displaying the peptides could be conducted to determine the apparent Kd of the derived particles. Fig. 2A may be moved to Fig S2 with less emphasis put on the conclusion that the apparent Kd of pep1osy compares favorably to GE11.
  2. I believe FC and microscopy experiments require a negative control, such as an unrelated peptide that can be expected not to bind to the cells. Without this, it’s hard to judge whether the results are due to specific peptide-EGFR or non-specific peptide-cell interactions.
  3. Fig 2C: It is not clear which signal results from the Lysotracker and which from the peptides. This should be represented in single images and as merged.

Minor:

  1. Can the authors please explain in a bit more detail how the 4 candidates were chosen from the initial top 30 solutions.
  2. Figure 1B-D: Might be useful to present the EGFR dimerization structure with a transparent surface so the docking can be evaluated in more detail. Also, the resolution of the structures is a bit low (this also applies to Fig 4 structures).
  3. Some peaks in Fig S23 and S24 (ESI MS spectra) are not labeled but should be for completeness.
  4. Figure S4A: It’s not entirely clear which bar represents 4C and which represents 37C.
  5. There is a problem with the font in the supplementary starting from Fig S5 caption.
  6. Fig S5: why are only 3 h for the control and 5 h for hEGF assessed?
  7. Line 317-319: ‘This analysis showed that 318 the putative folding of the peptide in the three-fold spike of the capsid does not necessarily correlate 319 with the previously generated free peptide models’ – this is not surprising as SWISS Model generates a homology model of the sequence based on the provided template. The peptide, not being part of the template structure, would have been modeled ab initio which most likely uses different constraints than PEP-FOLD 2.0.
  8. Line 433: Should be ‘primary secretory organ’
  9. Please check molar concentration units throughout the manuscript. Often given as xm which is confusing as this indicates length rather than concentration.
  10. Line 675: Reference is missing here.

Author Response

Reviewer 2

Major:

  1. It’s relatively hard to judge whether the apparent Kd for pep1osy can be derived from the fluorescence polarization assays. In my opinion, there are not enough datapoints to fit this curve reliably. The authors should consider using a complementary technique such as SPR or ITC to test the affinity of the peptides (characterisation of GE11 as described in ref 39 might be helpful in this respect). Alternatively, a phage ELISA with phage particles displaying the peptides could be conducted to determine the apparent Kd of the derived particles. Fig. 2A may be moved to Fig S2 with less emphasis put on the conclusion that the apparent Kd of pep1osy compares favorably to GE11.

* We agree that the fluorescent polarization data are only a rough estimate and moved the respective figure to the supplement (See Figure S2). We also adapted the main manuscript in this section to reflect lower prominence and stress the uncertainty of the measurement. We would not be able to provide more data point using the same method due to the limited solubility of the receptor and different methods are currently not available.

  1. I believe FC and microscopy experiments require a negative control, such as an unrelated peptide that can be expected not to bind to the cells. Without this, it’s hard to judge whether the results are due to specific peptide-EGFR or non-specific peptide-cell interactions.

* We believe that generally it is hard to find a suitable peptide negative control that can be used across all experiments. Initially, we considered the use of the linear precursor as a negative control in those experiments, but, due to potential oxidation and formation of a disulfide bridge under the given reaction conditions, this was not seen as a useful control. Some authors use scrambled peptides, but each version of these might have its own specific properties. This was the reason why we included pepEDA in our analysis. This peptide was already extensively analyzed in different publications and serves as a good benchmark. This peptide is promoted as ‘binder’ but the Kd is close to what researchers working with high affinity ligands would rank as unspecific interaction. Since we compare different peptides, which turn out to interact to different extends, and since we use complementary biophysical and cellular experiments to assess function, we believe that we provide a relevant picture of the performance of the peptides.

  1. Fig 2C: It is not clear which signal results from the Lysotracker and which from the peptides. This should be represented in single images and as merged.

* We thank the reviewer for pointing this out. We included the single images in the supplementary information (see Figure S5) to provide the full experimental details.

Minor:

  1. Can the authors please explain in a bit more detail how the 4 candidates were chosen from the initial top 30 solutions.

* We included the following statement to clarify the selection process: “The top 30 solutions were visually inspected for hydrogen bond satisfaction, burial of hydrophobic surfaces, and binding surface complementarity, and four favorable candidates by these criteria were chosen for additional rounds of high-resolution docking and sequence design.”

  1. Figure 1B-D: Might be useful to present the EGFR dimerization structure with a transparent surface so the docking can be evaluated in more detail. Also, the resolution of the structures is a bit low (this also applies to Fig 4 structures).

* We thank the reviewer for this suggestion. We changed the structure representations and increased the resolution in Figure 1 and Figure 4. Furthermore, we changed the structures in Figure 1 to give a more detailed overview on the peptide binding to the EGFR dimerization surface and included the transparent surface representation.

  1. Some peaks in Fig S23 and S24 (ESI MS spectra) are not labeled but should be for completeness.

* The labels have been added to those Figures to provide a complete overview on the measurements see Figure S24 and Figure S25 (the numbering has changes due to the additional microscopy images).

  1. Figure S4A: It’s not entirely clear which bar represents 4C and which represents 37C.

* We agree with the reviewer that the representation of shading in the pdf version was not clear. We improved the figures and the shading of the bars is now visible in our pdf export.

  1. There is a problem with the font in the supplementary starting from Fig S5 caption.

* Using our pdf export and Acrobat as a reader, we have not found the issues with the font in the captions. We double checked that all captions are formatted with the same style in word.

  1. Fig S5: why are only 3 h for the control and 5 h for hEGF assessed?

* In wound healing assays usually the first hours are the most important in the analysis. Since for the non-control samples more points were recorded, these were included in the evaluation. As can be seen from the figures, using only 3 h of the samples would not change the result.

  1. Line 317-319: ‘This analysis showed that 318 the putative folding of the peptide in the three-fold spike of the capsid does not necessarily correlate 319 with the previously generated free peptide models’ – this is not surprising as SWISS Model generates a homology model of the sequence based on the provided template. The peptide, not being part of the template structure, would have been modeled ab initio which most likely uses different constraints than PEP-FOLD 2.0.

* Apart from the elaborated peptide design, our structural modeling using preconfigures servers is a rather simplistic approach to get an idea of potential structures. We would not have assumed that structures do look the similar but just wanted to note that there are differences. We changed the phrasing in the sentence to avoid misunderstandings (see Line 319ff).

  1. Line 433: Should be ‘primary secretory organ’

* We corrected this typo. (Line 438)

  1. Please check molar concentration units throughout the manuscript. Often given as xm which is confusing as this indicates length rather than concentration.

* As often used in organic chemistry publication, we used a lower case small caps “m”. To be formatting and font independent, we changed this to a upper case standard font “M” throughout the manuscript.

  1. Line 675: Reference is missing here.

* We restored the link to Table 3.

Reviewer 3 Report

Assessing the overall risk posed by EGFR signaling induced cancer progression is indeed important, so I value this work.   All research questions are derived and sustained by the literature reviews and are pertinent to the proposed study. The paper is well written, with a clear structure and careful explanations throughout, enabling others to replicate these techniques if desired. The accuracy of results is convincing, and the conclusions appear to be reliable. To conclude, I recommend the publication of the current manuscript with the specific comments below, which authors are advised to address to strengthen their manuscript.

  • EGFR is known to regulate many crucial cellular programs in different pathological setting predominantly cancer progression, with seven different activating ligands influencing cell signaling in diverse ways. Each of its ligand has different properties confining to structural conformation changes before endocytosis of the receptor as the receptor dimerization conformational change has an impact on inside cellular signaling. Studies have been shown that EGFR dimerization depends on the binding ligand. For instance, some ligands do asymmetric dimerization and some results in symmetric dimerization (Wilson et al 2009 and Ronan et al 2016). The type of dimerization decides the fate of cellular signaling whether to differentiate or proliferate. Hence, I feel it is very important to know what exactly the pep1jhf and pep1osy are doing to the fate of EGFR dimerization and hence leading to its internalization and signaling. Fig 1 shows the design of peptides based on EGFR dimerization arm, still is it possible for authors to address whether the dimerization due to mentioned peptides leading to asymmetric/symmetric computationally or discuss this with known ligands function?
  • Does the alteration of cellular responses by pep1jhf and pep1osy are same or different with the sustained EGFR internalization? Discuss
  • Discuss the effect of pep1jhf and/or pep1osy in EGFR recycling vs degradation with relevant references pertaining to EGFR canonical ligand interaction.

Minor concerns:

  • Add appropriate statistical methods to justify significance of the results in appropriate figures (In particular, all biological functional assays).
  • Fig 2C: Change the image of pepEDA in Mcf7 cells. It has very few NucBlue positive cells compared to other two peptides which misleads the interpretation of results.

Ref:

Ronan T, Macdonald-Obermann JL, Huelsmann L, Bessman NJ, Naegle KM, Pike LJ. Different epidermal growth factor receptor (EGFR) agonists produce unique signatures for the recruitment of downstream signaling proteins. Journal of Biological Chemistry. 2016 Mar 11;291(11):5528-40.

Wilson KJ, Mill C, Lambert S, Buchman J, Wilson TR, Hernandez-Gordillo V, Gallo RM, Ades LM, Settleman J, Riese DJ. EGFR ligands exhibit functional differences in models of paracrine and autocrine signaling. Growth factors. 2012 Apr 1;30(2):107-16.

Author Response

Reviewer 3

Assessing the overall risk posed by EGFR signaling induced cancer progression is indeed important, so I value this work.   All research questions are derived and sustained by the literature reviews and are pertinent to the proposed study. The paper is well written, with a clear structure and careful explanations throughout, enabling others to replicate these techniques if desired. The accuracy of results is convincing, and the conclusions appear to be reliable. To conclude, I recommend the publication of the current manuscript with the specific comments below, which authors are advised to address to strengthen their manuscript.

  • EGFR is known to regulate many crucial cellular programs in different pathological setting predominantly cancer progression, with seven different activating ligands influencing cell signaling in diverse ways. Each of its ligand has different properties confining to structural conformation changes before endocytosis of the receptor as the receptor dimerization conformational change has an impact on inside cellular signaling. Studies have been shown that EGFR dimerization depends on the binding ligand. For instance, some ligands do asymmetric dimerization and some results in symmetric dimerization (Wilson et al 2009 and Ronan et al 2016). The type of dimerization decides the fate of cellular signaling whether to differentiate or proliferate. Hence, I feel it is very important to know what exactly the pep1jhf and pep1osy are doing to the fate of EGFR dimerization and hence leading to its internalization and signaling. Fig 1 shows the design of peptides based on EGFR dimerization arm, still is it possible for authors to address whether the dimerization due to mentioned peptides leading to asymmetric/symmetric computationally or discuss this with known ligands function?

* This comment highlights an important point and we included the literature as suggested into our introduction. The peptides described in this manuscript were predominantly designed for tumor and EGFR targeting without potentially activating EGFR. Peptides targeting the dimerization interface of EGFR obviously would upon intended binding prevent the known mode of receptor dimerization. The focus of our manuscript is peptide and AAV targeting to EGFR expressing cells. Next to the potential clustering of EGFR, AAV transduction likely is also facilitated by AAV specific secondary receptors such as the AAVR or other more serotype specific cell surface receptors. Computational studies of the effect of the viral presented peptides on EGFR clustering would be highly speculative due to the lack of data. Additional experimental studies with peptides on dendrimers or EGFR mutants would warrant an own publication. To this end, we did not study the effect of the peptides on the receptor signaling in more detail. However, this would be an interesting set of experiments that could be analyzed in the future to complete the current findings from a different perspective.

  • Does the alteration of cellular responses by pep1jhf and pep1osy are same or different with the sustained EGFR internalization? Discuss
  • Discuss the effect of pep1jhf and/or pep1osy in EGFR recycling vs degradation with relevant references pertaining to EGFR canonical ligand interaction.

* We value the reviewer’s comments on the cellular responses and also the receptor recycling. Again, we want to emphasize that the main aim of this manuscript is to describe peptides that are suitable for insertion into the AAV capsid to allow a retargeting towards tumor cells. The long-term goal is to enable a suicide gene therapy, which leads to the death of the EGFR overexpressing cell regardless of any effects on signaling. There are several published studies of pepEDA, which also targets the dimerization interface, and the detailed effects on receptor signaling remain elusive. In our A431 wound healing assays neither pepEDA nor pep1jhf at 5 µM significantly influenced the wound closure in the presence of 1 nM EGF. This suggests that these peptide on their own have little effect on EGFR signaling at typical therapeutic concentrations. The detailed intracellular processes are interesting, but at this point are not the topic of this study as the EGFR merely provides a docking point for the virus. We do agree that looking into and discussing the peptides’ effect on signaling is worthwhile to study. We included in the discussion of our manuscript that a deeper understanding of the peptides in generally is of importance to the field.

Minor concerns:

  • Add appropriate statistical methods to justify significance of the results in appropriate figures (In particular, all biological functional assays).

* We provide for all experiments the number of biological and technical replicates and represent the average and indicate errors in the respective figures. For the animal experiments we do use standard box-whisker plots, which also additionally include the mean. In this context we neither use the word ‘significant’ nor ‘significance’. In line with recent developments of data evaluation (Nature 567, 305-307 (2019); doi: https://doi.org/10.1038/d41586-019-00857-9 ; Nature 567, 283 (2019) doi: https://doi.org/10.1038/d41586-019-00874-8) we try to provide a more nuanced picture for example by testing various experimental approaches. The animal experiments show a broad variation as communicated by the error bars. We agree that, if we would make claims of treatment efficacy for our compound, we should provide a rigorous statistical analysis. However, due to the observed variations we would like to refrain from such claims and hence we wonder what we should justify by advanced statistical methods.

  • Fig 2C: Change the image of pepEDA in Mcf7 cells. It has very few NucBlue positive cells compared to other two peptides which misleads the interpretation of results.

* The image pepEDA in MCF7 cells shows 4 cells, which is similar to the number of A431 cells (which should not grow above 80% confluency) in the upper three images. In this light we feel that this number is sufficient to illustrate the localization within the cells. Larger number of cells were analyzed by flow cytometry as depicted in the first panel of the figure.

Ref:

Ronan T, Macdonald-Obermann JL, Huelsmann L, Bessman NJ, Naegle KM, Pike LJ. Different epidermal growth factor receptor (EGFR) agonists produce unique signatures for the recruitment of downstream signaling proteins. Journal of Biological Chemistry. 2016 Mar 11;291(11):5528-40.

Wilson KJ, Mill C, Lambert S, Buchman J, Wilson TR, Hernandez-Gordillo V, Gallo RM, Ades LM, Settleman J, Riese DJ. EGFR ligands exhibit functional differences in models of paracrine and autocrine signaling. Growth factors. 2012 Apr 1;30(2):107-16.

Round 2

Reviewer 2 Report

Thank you for addressing all comments. I will recommend this manuscript for publication.